# Self-Mixing Laser Distance-Sensor Enhanced by Multiple Modulation Waveforms

**DOI:** 10.3390/s22218456

**Published:** 2022-11-03

**Authors:** Federico Cavedo, Parisa Esmaili, Michele Norgia

**Affiliations:** Department of Electronics, Information and Bioengineering, Politecnico di Milano, 20133 Milan, Italy

**Keywords:** optical rangefinder, self-mixing interferometry, laser sensors, distance measurement

## Abstract

Optical rangefinders based on Self-Mixing Interferometry are widely described in literature, but not yet on the market as commercial instruments. The main reason is that it is relatively easy to propose new elaboration techniques and get results in controlled conditions, while it is very difficult to develop a reliable instrument. In this paper, we propose a laser distance sensor with improved reliability, realized through a wavelength modulation at a different frequency, able to decorrelate single measurement errors and obtain improvement by averages. A dedicated software is implemented to automatically calculate the modulation pre-emphasis, needed to linearize the wavelength modulation. Finally, data selection algorithms allow to overcome signal fading problems due to the speckle effect. A prototype demonstrates the approach with about 0.1 mm accuracy up to 2 m of distance at 200 measurements per second.

## 1. Introduction

Due to high resolution and non-invasive features, optical distance sensors are one of the most used techniques for accurately estimating the absolute distance of a remote target. Depending on the application, these rangefinders can be divided in three main techniques, known as laser triangulation, Time-of-Flight (ToF), and absolute interferometry. In laser triangulators, the absolute distance is determined by measuring the viewing angle between a source and a receiver [1]. Such triangulators are more suitable for short distance measurements and they can reach micrometer resolution over a few millimeters [2]. However, they are limited for sensing inside objects through narrow openings. This is mainly because of triangular geometry which implies the minimum width of the sensor to provide the distance between its transmitter and receiver. For long distances, instruments based on the Time-of-Flight technique show reliable performance with resolution between 1 mm and 1 cm [3,4] and are extensively employed for commercial and consumer applications. Distance measurement systems based on interferometric techniques have been studied extensively, due to demanding higher resolution in industrial and scientific environments. Despite high accuracy, costly and complicated optical setup limits the industrial applications of such techniques [2]. Alternatively, self-mixing interference effect in a laser diode [5,6,7] has led to widespread applications of this technique for measuring relatively short distances with high resolution while remaining very small in size and cost-effective. Due to its advantages, the application of sensors based on the self-mixing technique are not limited only to absolute distance measurement. Various applications are reported in literature, including displacement [8,9,10,11,12], speed [13,14] vibrations [15,16,17,18,19,20], angles [21], imaging [22], liquid flow and level [23,24,25,26,27], biomedical applications [28,29,30], in addition to laser parameters measurements [31,32] and measurement for laser ablation [33,34]. Self-Mixing Interferometry (SMI) takes advantage of the optical back-injection inside the laser cavity. It occurs when a fraction of light emitted by the laser diode is reflected by an external target back to the laser cavity, and mixes with the internal lasing field. Consequently, it causes modulation of both laser frequency and emitted power, which gives information about the target position [5,6,7]. Figure 1 shows a very-simple setup for SMI: it consists of the laser diode, photodiode inside the laser cavity (known as monitor photodiode, normally inside the laser case), a collimating lens, and target. The monitor photodiode directly measures the power emitted by the laser diode, *P(ϕ)* which is a periodic function of the back-injected phase, *ϕ = 4π·s/λ*, where *λ* is wavelength of the laser and *s* is the distance from the laser to the target [5].

For estimating absolute distance from the target, the standard approach consists in modulating the laser wavelength and measuring the fringes period. The very-first signal processing was a simple fringes-counting [35]. A more effective approach to estimate the absolute distance is to measure the fringes period in the frequency domain [36]. Considering the approximate linear relationship between beat frequency and distance, different techniques can be applied to calculate beat frequency of fringes and then estimate the distance. In terms of the tradeoff between accuracy and elaboration time, interpolated Fast Fourier Transform (FFT) [37] provides a good approach to calculate beat frequency of fringes and consequently estimate the distance [38]. However, interpolated FFT exhibits limits if signals are not perfectly sinusoidal, in that case, the interpolation formula is not valid anymore. The interpolation errors have the strong disadvantage of being systematic and therefore they cannot be reduced by averaging operations. Different approaches are proposed in literature to improve the accuracy in fringe frequency estimation. A direct evolution of interpolated FFT is all-phase FFT [39], proposed to reduce the influence of spectrum leakage and signal noise, at the expense, however, of a longer processing window and a consequent increase in execution time. Furthermore, this technique is incapable of compensating for error due to the lack of perfectly constant frequency as in real self-mixing signals. An algorithm based on MUltiple SIgnal Classification (MUSIC) is also proposed for frequency estimation in the SMI-based distance and velocity sensing system [40]. In general, the MUSIC method assumes that a signal vector consists of a known number of complex exponentials, whose frequencies are unknown. The proposed MUSIC algorithm-based SMI distance sensing system achieved a better Signal-to-Noise Ratio (SNR), with respect to FFT in a range from 20 to 100 cm. Although the performance advantages of MUSIC are remarkable, such technique requires extensive computation effort to be executed in real time. Another approach, based on the Genetic Algorithm (GA) is proposed in [41]. Cost function is established based on emitted power variation from the linearly modulated laser diode. To overcome premature convergence and time complexity of the GA, an improved GA algorithm based on the father–offspring combined selection method [42], in addition to a shrinking exploration range approach [43], is proposed in [41]. Using an 850 nm vertical cavity surface emitting laser (VCSEL), the absolute distance is estimated in the range from 2.4 cm to 20.4 cm with 10 μm resolution. Despite high resolution in the distance measurement and improvement in selection and exploration range shrinking compared with the original GA, the speckle effect is neglected by assuming greater longitudinal speckle size to compare with an equivalent displacement in the whole measuring range. The random phase superposition of the back-reflected light arising from diffusive targets, known as speckle effect, causes amplitude fading of the SMI signal. This deteriorates robustness of the GA-based SMI sensing system since the self-mixing signal, which exhibits amplitude fading, strongly affects the cost function and leads to wrong convergence of the distance value.

In this paper, a technique based on interpolated FFT with multiple modulation is proposed to overcome limits in self-mixing rangefinders, by decorrelating determinist error sources. A software able to automatically calculate the optimum modulation pre-emphasis is developed, and an algorithm based on amplitude of lateral bins is applied in addition, to reduce the error induced by the speckle effect. A real-time prototype demonstrates the effectiveness of the proposed approach. The realized distance sensor is characterized up to 2 m using a grating ruler, getting a maximum nonlinearity error lower than 0.4 mm and standard deviation limited to 0.1 mm. 

The rest of the paper is organized as follows: In Section 2, the SMI distance measurement and non-linearity in modulation are discussed. Section 3 describes the realized prototype and Section 4 investigates possible solutions to address deterministic errors and introduces the proposed multiple modulation technique. Experimental results as well as the performance evaluation are reported in Section 5 for both distance and vibration measurements. Finally, the conclusion and final marks are presented in Section 6.

## 2. Distance Measurement with Self-Mixing Interferometry

The measurement of absolute distance with self-mixing interferometry is commonly realized through a wavelength modulation [38]. When the target is at rest, the laser wavelength modulation induces a shift of the interferometric phase *ϕ* = 4π·*s*/*λ*, proportional to the distance *s*:(1)$$\frac{\partial \varphi }{\partial \lambda }=-4\pi \frac{s}{\lambda^{2}}$$

The easiest way for modulating the laser wavelength is acting on the pump current *I*. The evaluation of the fringe frequency *f_tone_*, during the modulation, provides a measurement of the target distance:(2)$$s=-\frac{\lambda^{2}}{2\cdot \left(\frac{\partial \lambda }{\partial I}\right)\cdot \left(\frac{\partial I}{\partial t}\right)}f_{tone}$$

The standard modulation shape is triangular [23], and the distance value is obtained by averaging the frequencies of the ascendant and descendant phases of the triangular wave. This average improves the accuracy and cancels the contribution of a possible target motion [44]. Function (2) is linear with *f_tone_* only for small modulation amplitude and for low-frequency modulation, because the factor (∂*λ*/∂*I*) is not constant with *I* and should be compensated. Indeed, there are two reasons for non-linearity: the dependence of (∂*λ*/∂*I*) on *I* [45] and its frequency dependence [46], mainly due to the thermal behavior of the laser diode. Figure 2 shows the measurement of (∂*λ*/∂*I*) for the DFB laser at 1550 nm which will be used in the realized prototype (model WSLD-1550-020m-1-PD).

In telecommunications, pre-emphasis refers to a system process designed to emphasize high-frequency signal components and improve the overall Signal-to-Noise Ratio (SNR) to therefore minimize the adverse effects of attenuation distortion. A pre-emphasis of the modulation curve could compensate for the non-linearity, but the correct distortion shape cannot be retrieved by a simple analytic model, because it is a function both of modulation amplitude and frequency, and of repetition rate of the modulating curve. The reasons are the particular behavior of the non-linearity (Figure 2, as example), and the thermal response of the LD, which is described in frequency by several poles-zeros and involves a long-time behavior [46].

## 3. Realized Prototype

A prototype of a self-mixing rangefinder was developed, composed of two parts: the analog electronics necessary to adapt the fringes signal and to inject a particular current waveform into the laser diode, and a commercial data acquisition card (DAQ, Analog Discovery II, from Digilent, Seattle, WA, USA) for generating the modulating wave and acquiring the signal (the conceptual scheme is the same as in [38]). The block scheme of the instrument is shown in Figure 3. The laser driver was realized by a standard current generator, composed of an Operational Amplifier (OPA) and a MOSFET transistor with feedback on the tail resistor. The OPA bandwidth was 20 MHz and the modulation signal was limited by a second-order low pass filter, to filter out DAC spurious harmonicas.

Analog Discovery 2 includes a 14-bit, 30 MHz Analog-to-Digital Converter (ADC), and a 14-bit 10 MHz Digital-to-Analog Converter (DAC), both with a maximum rate (acquisition/generation) of 100 MSPS. Through the DAQ, it is possible to generate an arbitrary waveform for the laser modulation and acquire synchronously the interferometric signal. As expected, a pure triangular modulation induces fringes with variable frequency, even on a target at rest. In order to accurately evaluate the wavelength modulation non-linearity, we developed a real-time software able to locally measure the fringe frequency. It is based on interpolated FFT calculated on a sliding window of 32 samples, following the approach proposed in [47]. Considering the sampling frequency of 10 MSPS, 32 points are enough to acquire some fringes to elaborate the tone estimation, without losing too much resolution on the local frequency measurement. We repeat this operation 10 times on different acquisitions and we take the average of all obtained traces. Figure 4 shows the interferometric signal before and after the pre-distortion. Figure 5 shows the corresponding measurements of the fringes period: for a pure triangular modulation at 10 kHz, and for the pre-distorted wave.

Due to edge effects at the beginning of the triangular wave, the fringes period cannot be made constant throughout the whole modulation period, but it is enough for the next elaboration to realize a flat interval of sufficient length. Great care should be taken to set the interval size, in order to get a useful number of samples, for example, 256 or 512, for optimizing FFT elaboration and measurement speed.

Once the modulation shape has been optimized through pre-distortion, the frequency of the fringes remains constant during the modulation, also at different target positions, and increases linearly with the distance, as expected. Figure 6 shows some examples of self-mixing signals for different target distances (25 cm, 60 cm, and 100 cm). The upper panel also reports the corresponding modulating wave at 10 kHz with respect to a triangular wave: it is evident that the non-linearity is different on the two fronts. This is due to the sum of two effects, nonlinearity and frequency response, which add up in one phase and subtract in the other. The signal under elaboration is given by the self-mixing signal after the subtraction of the modulating wave [38] and high-pass filtering. It should be noted that the signal amplitude is not constant: as expected it is a function of the laser current (higher at higher pump current). Even after having linearized the wavelength modulation very well, better than 10^−3^, there are still periodic systematic errors with distance, also depending on the Signal-to-Noise Ratio. This kind of error is deterministic and cannot be reduced by averaging.

## 4. Methods to Overcome Deterministic Errors

A deep study of the error sources found that they are mainly due to the limit of the interpolated FFT for signals not perfectly sinusoidal and with noise, as reported in [39] and further described in [48]. In order to fix this problem, we studied a more complex modulation scheme, composed of multiple modulating waveforms, at slightly different frequencies. The modulating waves are designed in order to realize different fringes frequencies. In particular, it is relevant to have different frequencies positions with respect to the FFT bins, because the interpolated-FFT technique shows deterministic errors depending on that position. When one waveform induces a fringes frequency exactly over a bin, the others modulating waveforms generate frequencies between bins. In this way, the deterministic errors due to interpolated FFT are different for every waveform and, more importantly, they are not correlated to each other. In this way, we get three frequency measurements, uncorrelated, with also the possibility of discarding one if coming from a bad position in frequency, where the interpolated FFT error is maximum. The worst position is exactly over a bin of the FFT, because in that position, the two-bin interpolated FFT on a real signal shows the maximum error.

The proposed technique is a good way to decorrelate single measurements, obtaining improvement from an average procedure. The limit, however, of this approach is the very long procedure to generate the correct distortion manually, because individual waveforms influence each other: a distortion applied on the first modulating wave (at 10 kHz) changes the frequency measurement also for the following waveforms. Furthermore, once the optimal distortion has been found for one waveform (for example, at 10 kHz), it cannot be used for the others modulating waves through a simple re-sizing. For this reason, the pre-emphasis procedure must be carried out simultaneously on the entire modulating signal. 

To speed up the process, we implemented a recursive algorithm on LabVIEW, able to automatically optimize the modulation shape. In a first approximation, the modulating curve should be proportional to the integral of the measured fringes period (Figure 4), given by a pure triangular modulation. Experimental evidence demonstrates that the curve thus obtained provides overcompensation, and it is impossible to find a closed formula for getting the correct pre-emphasis, because the thermal response depends also on the previous modulating waves. Our solution consists in implementing an iterative procedure for generating the modulating wave. It starts with a pure triangular modulation, and measurement of fringes period. The next modulation wave is obtained by the first one, adding a corrective wave (a pre-emphasis). The pre-emphasis is calculated as 90% of the theoretical one, given by the integral of the measured fringes period minus the target value. In the second step, the system applies the new modulating wave and measures again the resulting fringes period. A third modulating curve is then calculated by adding again the new pre-emphasis calculated as in the previous step. That operation is repeated for a few times (typically 5–10), until reaching a relative maximum error in fringes frequency lower than 10^−3^. The factor “90%” was evaluated empirically, finding the faster convergence to stable flat values for the fringes period. The typical duration of the whole pre-emphasis procedure is about 2 s in the automated procedure. This procedure is required only one time when the instrument is realized (once in the instrument’s life), there is no need to repeat it. During this procedure, the target should be at rest, because a target movement induces an error in the local frequency measurement.

Figure 7 shows a screenshot of the LabVIEW software able to automatically calculate the distortion of three triangular waves at different frequencies (10 kHz, 9.5 kHz, and 9 kHz), in order to linearize the fringe frequency in the measurement intervals. Figure 7 shows the self-mixing signal, the measured fringe frequency, and the original fringe frequency (before compensation), and also the intervals where the interpolated FFT is evaluated (vertical lines). Figure 8 shows the corresponding modulating wave, realized by this procedure, compared with the initial triangular wave. This modulating wave drives the laser current generator.

## 5. Measurement Results

For the metrological characterization of the proposed instrument, a mechanical setup was realized with a slit and a grating ruler (YH-5 um-1000), used as reference for distance measurements as shown in Figure 9. Three triangular waveforms at different frequencies (10 kHz, 9.5 kHz, and 9 kHz) with peak-to-peak amplitude of 20 mA are generated by DAC, with an update rate of 10 MSPS. This modulation signal is applied to the bias current of 70 mA. For each signal, 1000 samples are acquired at 10 MSPS. The DAQ card that manages the interferometer also acquires in real time the position of the target from the grating ruler, with an accuracy of 10 µm. This performance is adequate for the application because the target accuracy of the interferometer is one order of magnitude higher. It should be noted that all the measurements have been performed at room temperature when the laser diode reached its thermally stable regime.

### 5.1. Absolute Distance Measurement

A measurement campaign was conducted in order to evaluate instrument standard deviation and non-linearity. Figure 10 shows the measured frequency for each edge of the three modulating waves, at 9 kHz, 9.5 kHz, and 10 kHz, as a function of target distance. 

For every single ramp, ascending or descending, the fringes frequency shows linear dependence with distance, with slope proportional to the waveform frequency. This confirms a good optimization of the instrument response, realized by the modulation pre-distortion. By a linear combination of the measurements from the three different modulating waves, we get the absolute target distance: a regression curve is calculated for every curve of Figure 10, and the final results are given by the average of the six measurements, with the possibility to discard single data for improving the final accuracy. The causes of exclusion implemented are two: (i) too low tone amplitude (close to the noise floor); (ii) amplitude of the second highest bin of the interpolated FFT too low with respect to the highest one. Condition (i) is quite obvious, because a measurement point confused in the noise floor adds just errors; while condition (ii) is useful for discarding the measurement values in correspondence with the maximum error of the interpolated FFT. As explained in [37,38], the interpolation in frequency is realized between the two bins with higher amplitude in the FFT. When the tone is close to a bin, the second bin becomes very low, and can be covered by the noise floor. In this case, the lower the Signal-to-Noise Ratio, the more error the interpolation procedure adds, with respect to the real tone frequency.

The final prototype is able to measure up to 4 m of distance, with a native measurement rate of 2 kHz, realized through three modulating waves around 10 kHz. It was characterized by measuring different distances on the grating ruler, up to 2 m of distance. For every distance, the characterization software acquires the measurement of the grating ruler, and 200 measurements of the optical prototype, in native format (measurement rate 2 kHz) and after 10 averages (measurement rate 200 Hz). The double measurement, with and without averages, helps to understand the behavior of the instrument, in terms of repeatability. In the case of additive random noise, not correlated, we expect a factor √10 of improvement in standard deviation for the average on 10 samples. If the improvement is higher than the square root of the average number, typically some disturbances are present. 

The laser beam is focused at about 4 m; therefore, it is close to be collimated in the measurement range, considering the collimating lens diameter of 11 mm.

Figure 11 shows the linear behavior of the calibrated optical instrument over all tested ranges, without changing the optical condition (no autofocus or speckle-tracking techniques are applied [49]). 

Figure 12 and Figure 13 show the relative and absolute standard deviation, evaluated over 200 data, for the output at 2 kHz (no averages) and 200 Hz (10 averages). As expected, there is an improvement of about a factor of 3 for the averaged measurements.

In order to estimate the accuracy of the laser sensor, the absolute error of the mean distance (averaged over 200 data) has been evaluated as a difference with respect to the grating ruler measurement. This difference is an estimation of the nonlinearity of the optical sensor, in the hypotheses that the ruler has a better accuracy and that the target does not shift on the ruler. It should be considered that the average over 200 data still includes some tens of micrometers of variability due to the native standard deviation. In conclusion, Figure 14 shows a worst-case non-linearity of the laser sensor as a function of the distance. It is worth to note that, to our knowledge, an absolute accuracy has never been reported for a self-mixing range finder working in real-time on such a large measurement range.

There is an improvement also with respect to speckle effect, due to the possibility of discarding wrong measurements (with too low signal) in the repeated measurement, but also inside a single measurement composed by three modulating waves. We have experimentally seen that on a real aluminum target, the measurement is practically always running, for target distances up to 2 m.

### 5.2. Vibration Measurement

The prototype is also able to measure target vibration, by calculating the Doppler shift between the rising and falling edges of every modulating wave, as proposed in [44]. The sensitivity of the vibrometer is three orders of magnitude better with respect to absolute distance measurement: the measurement resolution becomes about 100 nm, with respect to 0.1 mm of the distance measurement. The Doppler effect allows to measure vibration with micrometric amplitude, as shown by an example of measurement reported in Figure 15. The target is a loudspeaker vibrating at 25 Hz with an amplitude of about 4 µm, placed at a distance of about 46 cm. The main limit in vibration measurements are the vibration frequency and the target speed. The limit in vibration frequency is given by the measurement rate of the instrument, 2 kHz, therefore we can reconstruct up to 1 kHz. The second limit is the target speed, which should not overcome the equivalent modulation speed: on one side of the modulation wave, we get a reduction in the number of fringes, while on the other side, we get an increase. For example, at 40 cm of target distance, we get fringes frequency *f* ≅ 1 MHz (Figure 10). In that case, the maximum measurable target speed is *f × λ*/2 ≅ 77 cm/s.

Additionally, this measurement is obtained from the average of the three modulations at different frequencies. The error correction procedure of the interpolated FFT is absolutely relevant for this type of measurement, especially for low-speed movements. In vibration measurements, a minimum imbalance error between the ascent and descent phases of the modulating wave results in a constant speed offset, hence, in a drift of the measured displacement. Moreover, for small vibration amplitudes, the slightest error in the interpolated FFT, especially while the target direction changes, leads to steps in the displacement measurement. Experimentally, there has been a notable improvement in the quality of the vibration measurement obtained with this technique, following the techniques described in Section 4. For example, in [44], the same technique was used for measuring target speed, with good results. However, in that case, the target was moving at a quite high constant speed (it was a rotating cylinder at about 40 cm/s), not vibrating, and in that condition, a small error in the frequency estimation does not lead to a relevant relative error on speed measurement.

## 6. Conclusions

In this work, we presented a novel approach to the laser self-mixing rangefinder, in order to overcome the limits that until now have not allowed this technique to become commercial. The requirements are: ability to work easily on a diffusing surface; good absolute accuracy (not just resolution or standard deviation); good reliability; high measurement speed; simple setup and automatic optimization, without requiring a dedicated researcher for the calibration. The proposed approach is based on multiple current modulation, with automatic setup of the pre-emphasis needed to linearize the wavelength modulation. This technique allows to obtain, over a range of 2 m, a nonlinearity comparable to the standard deviation of the measurement at 200 Hz (about 0.1 mm). In addition, there is the possibility to discard wrong measurements from the average (too low signal level or wrong position for the interpolated FFT), improving accuracy and robustness against local signal fading. These features make it possible to pass from a proposal in scientific literature to a real commercial tool, for several applications, such as 3D dimensional measurements, also through holes, or object localization. 

The actual work in progress is the development of a prototype entirely realized in embedded electronics, easy to reproduce in series, with some improvements: laser thermostat, for long-term stability; autofocus and/or speckle-tracking system for improving the probability of the useful signal for the measurement.

## Figures and Tables

**Figure 1 sensors-22-08456-f001:**
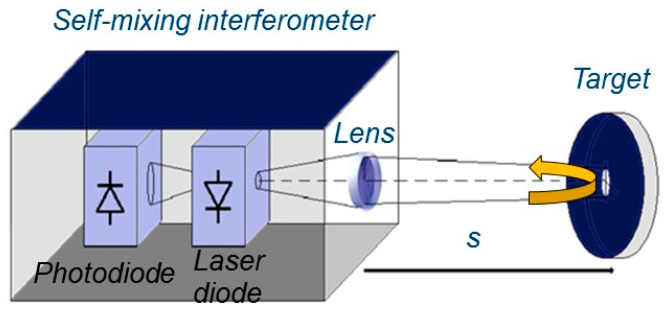
Standard setup for a self-mixing interferometer.

**Figure 2 sensors-22-08456-f002:**
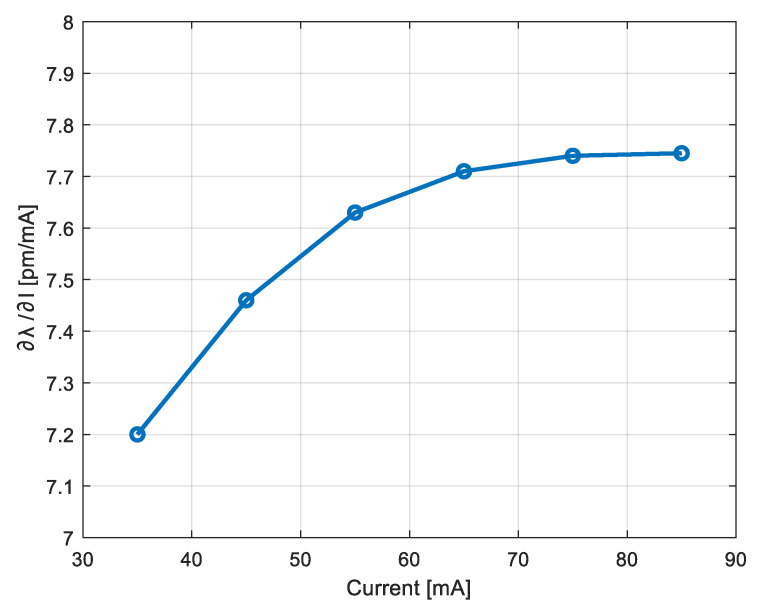
Measurement of (∂*λ*/∂*I*) for the DFB laser WSLD-1550-020m-1-PD, as a function of the pump current.

**Figure 3 sensors-22-08456-f003:**
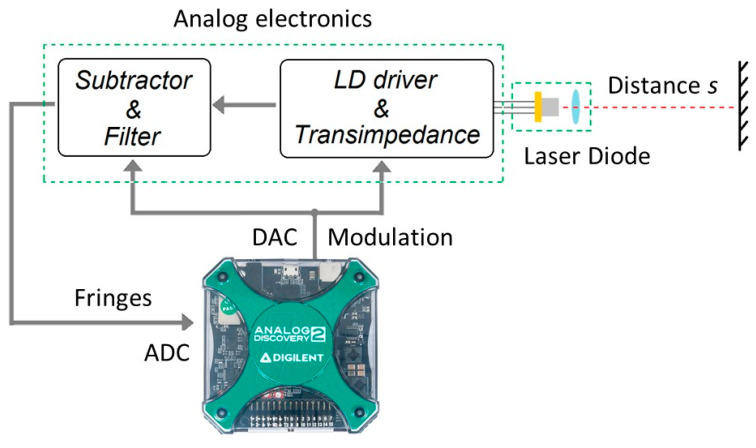
Block scheme of the self-mixing instrument.

**Figure 4 sensors-22-08456-f004:**
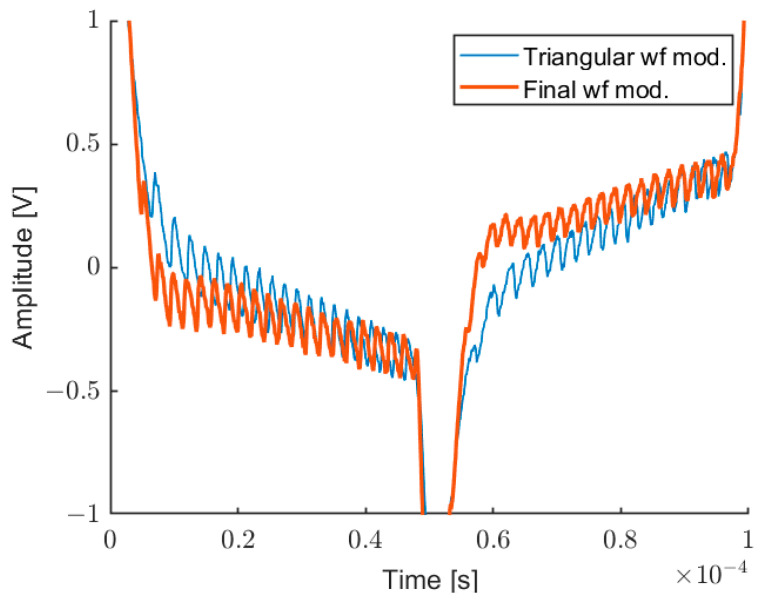
Interferometric signal before (thin line) and after (thick line) the linearity compensation.

**Figure 5 sensors-22-08456-f005:**
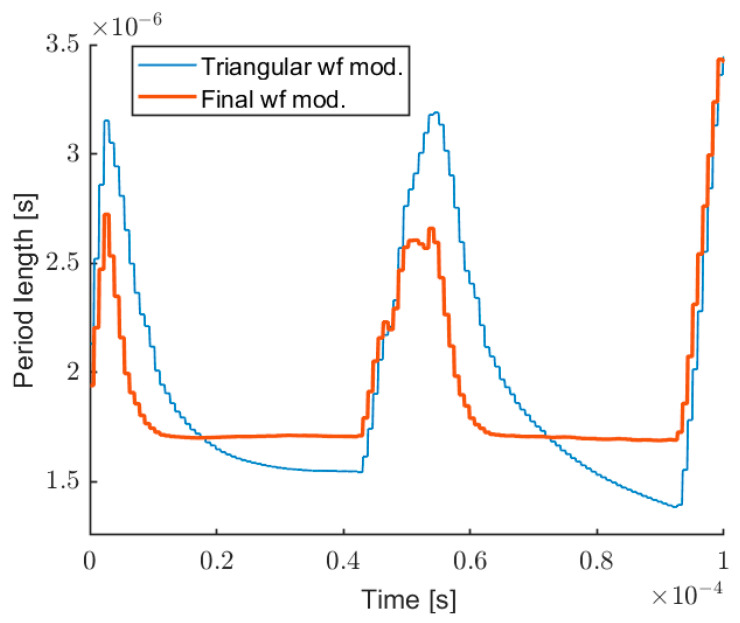
Period of the fringes before (thin line) and after (thick line) the distortion of the modulating wave.

**Figure 6 sensors-22-08456-f006:**
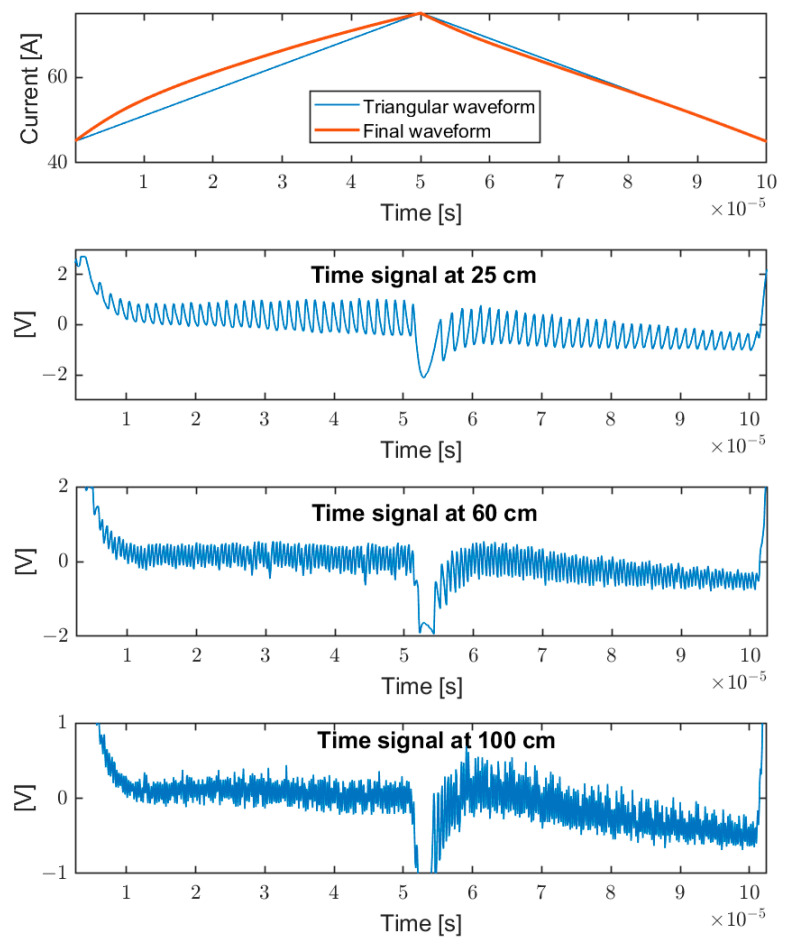
Upper pane: laser diode modulation, triangular (thin line), and pre-distorted wave (thick line). Lower panes: corresponding self-mixing signals, for different target distance (25 cm, 60 cm, and 100 cm).

**Figure 7 sensors-22-08456-f007:**
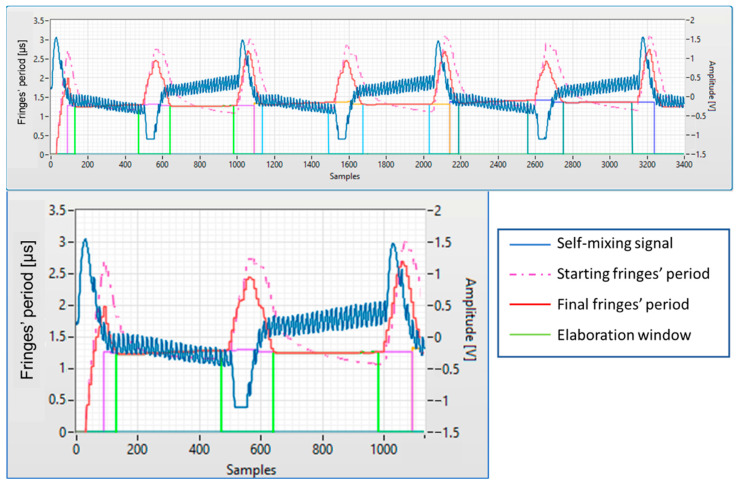
Top panel: a screenshot of the LabVIEW program for calculating the modulation distortions, showing the self-mixing signal, the measured fringes’ frequency, and the original fringes’ frequency (dash-dotted line). Vertical lines indicate the intervals for interpolated FFT execution (elaboration window). Bottom panel: detail of the first 10 kHz modulation.

**Figure 8 sensors-22-08456-f008:**
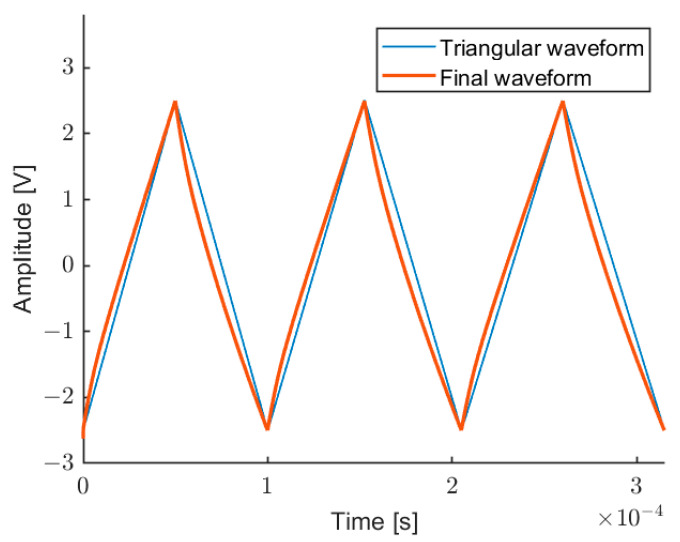
Output of the DAC, controlling the laser current generator, composed of waveforms at three frequencies (10 kHz, 9.5 kHz, and 9 kHz), after the pre-emphasis procedure (thick line) for getting a linear wavelength modulation, in comparison with the initial triangular modulation (thin line).

**Figure 9 sensors-22-08456-f009:**
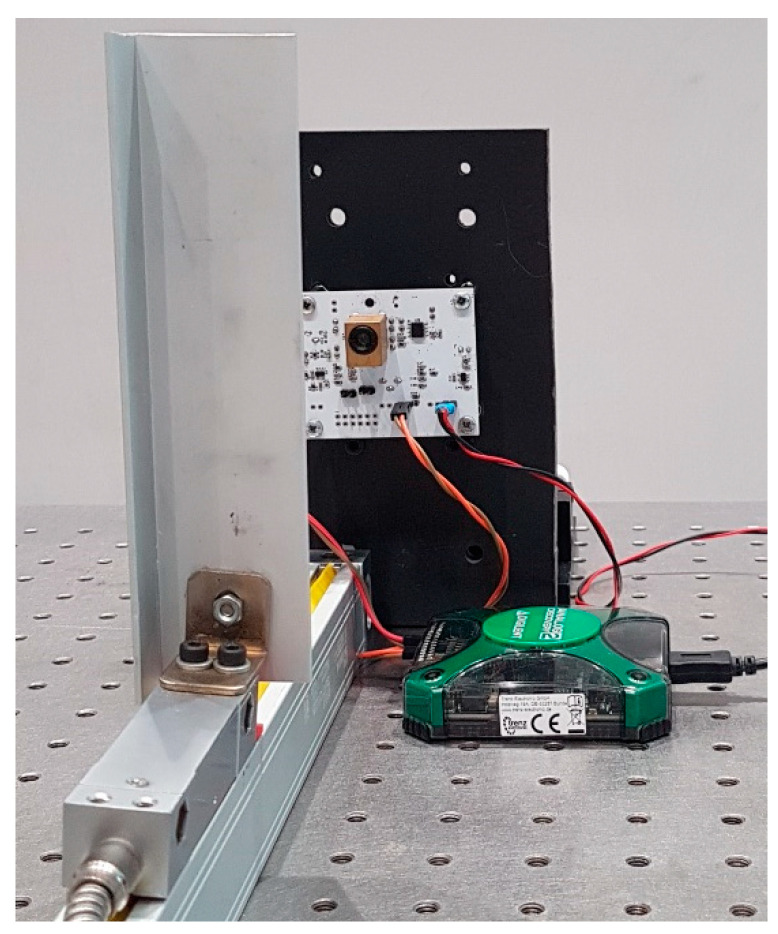
Experimental setup.

**Figure 10 sensors-22-08456-f010:**
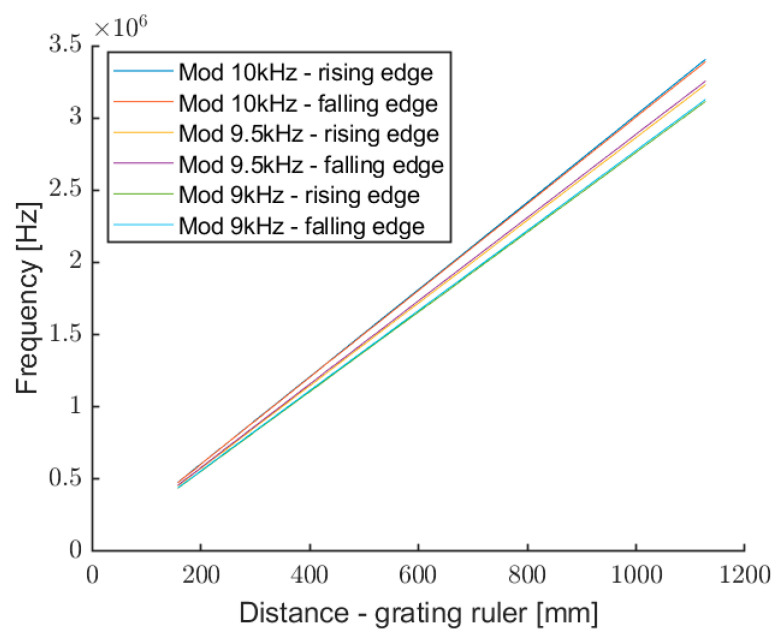
Frequency of the self-mixing signal as a function of the target distance, for rising and falling edges of the three modulation waves.

**Figure 11 sensors-22-08456-f011:**
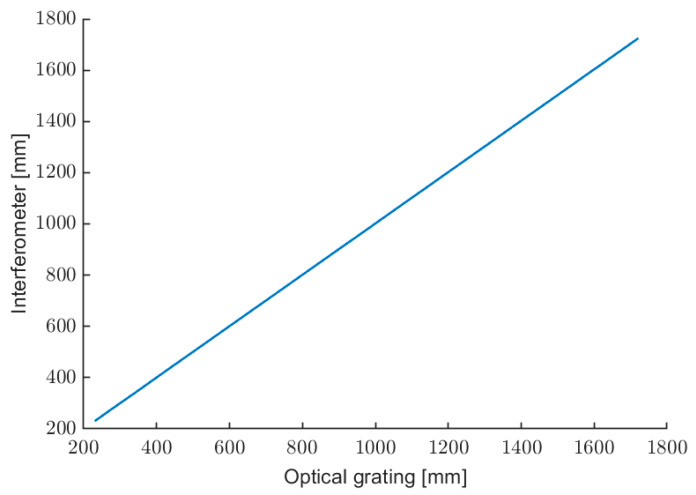
Characterization of the optical rangefinder.

**Figure 12 sensors-22-08456-f012:**
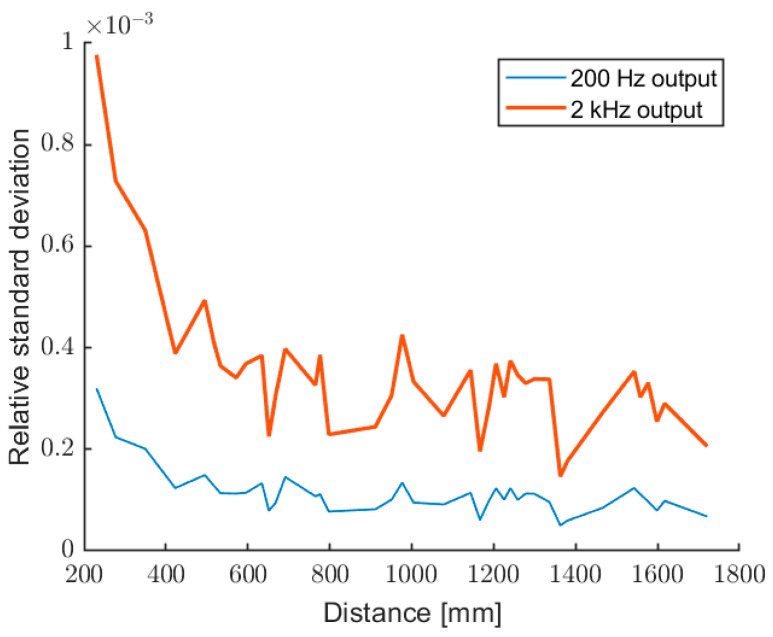
Relative standard deviation of the measured distance, for the outputs at 2 kHz (upper trace) and 200 Hz (after 10 averages).

**Figure 13 sensors-22-08456-f013:**
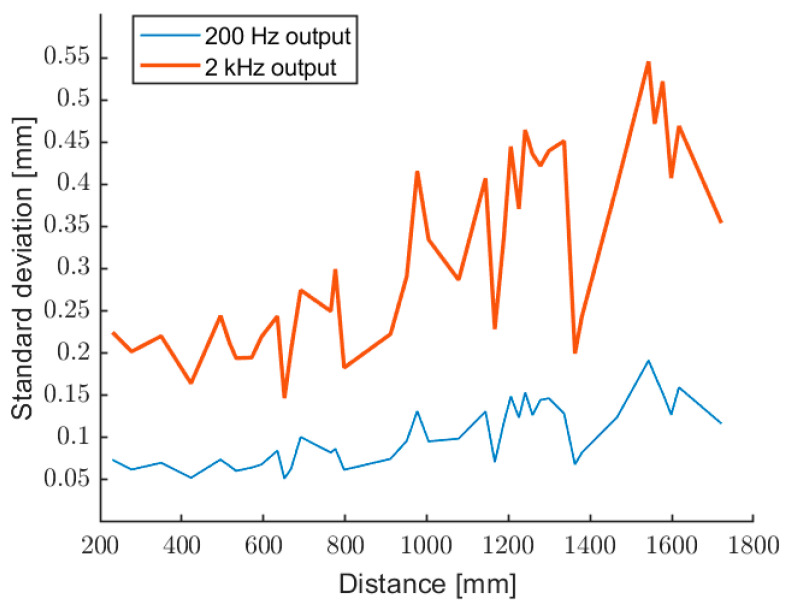
Absolute standard deviation of the measured distance, for the outputs at 2 kHz (upper trace) and 200 Hz (10 averages).

**Figure 14 sensors-22-08456-f014:**
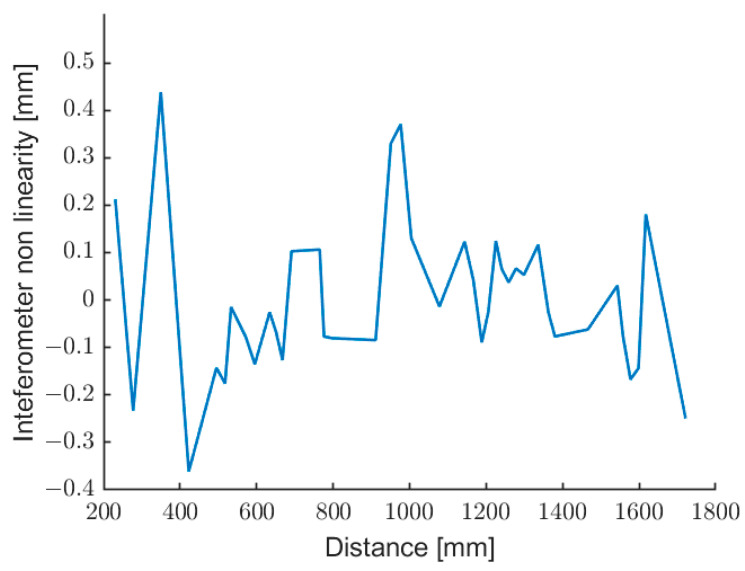
Difference between the measurements of the grating ruler and the optical sensor.

**Figure 15 sensors-22-08456-f015:**
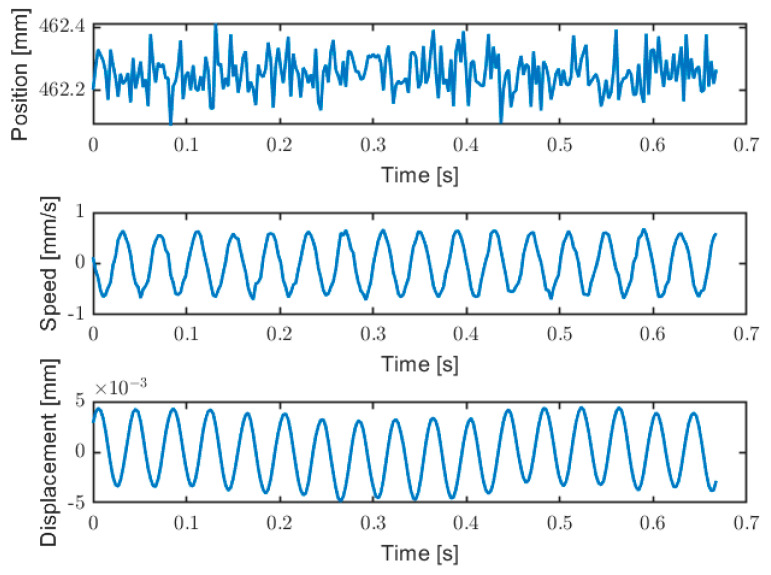
Example of vibration measurement. Upper trace is the absolute distance measurement; middle trace is the measured target speed (Doppler shift); lower trace is the reconstructed target vibration, by integration of the speed measurement.

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
