# Peer review of "Self-Mixing Laser Distance-Sensor Enhanced by Multiple Modulation Waveforms"

_sensors, 2022, doi:10.3390/s22218456_

Round 1

Reviewer 1 Report

The article presents an optimization work for the self-mixing interferometry technique for measuring absolute distances. The enhancement is mainly related to improvements in the algorithm that manages the laser diode modulation and the fringe detection. Experimental characterizations are presented. The article is suitable for the topics of the journal and the Special Issue. The general quality of the paper is good, and the results are interesting from the scientific point of view and its applications. However some parts are not clear and can be enhanced. I suggest a few revisions to improve the text readability and clarity, while providing more details on the presented system: once applied the article can be published.

Main comments:

A figure of the SMI measurement accuracy and main results of its characterization should be presented in the final part of the introduction. Presenting (only) the grating ruler accuracy is not useful and it can result in misleading.

Some experimental details that can be useful for understanding the measurements should be provided. For example:
Is the LD operated at constant temperature? At which temperature has been performed the characterization of Fig 2?

Which are the DAQ parameters, such as resolution and bandwidth? In particular, which ADC/DAC sampling rates are used?

Which LD driver is used, and which are its parameters? Could the bandwidth and linearity of the driver affect the modulation?

How “the non-linearity of the modulation has been compensated” at line 166? Is it an offline linearization using the calibration curve?

Could you provide more details regarding the modulation waveform? In particular, which are the actual current range and offset used for the LD current triangular wave?

The first paragraph of section 4 is not clear (and in particular lines 193...196). The different waveforms are modulated sequentially, or in a mixed way?

Figure 7 is not clear. I think that a different way of presentation should be found, with a proper legend for the different curves and intervals. Even a conceptual time diagram could be useful, to show unambiguously the temporal sequence, due to its importance in the work.

The convergence of the optimization is reached in “few times”. How many? Which is the typical (both absolute and relative to the measurement cycle) duration of the pre-emphasis? Does it have to be repeated each time, or just once per measurement session? Which is the role of the eventual target movement during the procedure?

Which is the relation between Figure 8 and the linearization of Figure 6? Which is the origin of the discretization observed in Figure 8?

The final prototype has a measurement rate of 2kHz. But what about the modulation speed? Is it still at ~10kHz?

The section regarding the vibration measurement seems out of context. If necessary for the current article, it should be introduced better, with adequate experimental details and references.
The sentence “The sensitivity as vibrometer is three order of magnitude better with respect to absolute distance measurement” should be better motivated.

Minor issues or suggestions:

The speckle problem is claimed to be overcome in the abstract. However this improvement is not evident from the results presented in the main text, this aspect can be highlighted better.

I think that Optical Coherence Tomography could be presented as a possible specific interferometric alternative technique for absolute ranging, since recently it is finding several commercial applications.

Among the presented SMI applications, also plasma concentration measurement in laser ablation has been proposed, as well as other industrial laser applications.

The MUSIC and GA algorithms are described extensively in the introduction, although they are not directly used in the work, with some unnecessary detailed results. This part can be simplified to highlight just the differences and improvements of the current algorithm.

Please, introduce the “pre-emphasis” technical term at its first usage.

Why “As expected, a pure triangular modulation induces fringes with variable frequency, even on a target at rest”?

I suppose that the “sliding window of 32 samples” depends on the ADC sampling rate. Please could you specify such a parameter?

Following a logical presentation, it seems like Figure 5 should be presented and discussed before Figure 4 (which is the consequence of Figure 5, if I understood well). A brief sentence to introduce the principles of the algorithm used to determine the fringe period could be added here.

A legend can be added to Fig 6 (upper plot).

Can you provide more details regarding the grating ruler used for the characterization? (model or characteristics, DAQ interface and measurement method)

Regarding figure 10 and 11, could you show the actual experimental points, not just the straight lines? Could you attribute error bars to them? Maybe a grid would also help in reading the plots.

I also suggest to uniform the plotting styles (brackets for units; “1” not as “unit” of pure numbers).

The article is well written, however few minor language errors are present (e.g. “the easier”… maybe “the easiest”?; “Furthermore, this technique is not however”; “the signal frequencies following frequency estimation”...). I suggest a general check.

Some curiosities:

Could you suggest some specific commercial or industrial application where the instrument might be applied?

From the text the algorithm is running onto a computer. Could you estimate its computational cost? For example, by providing the minimum resources that are required. In view of a commercial application, could it be implemented on a microcontrolloer or a FPGA?

Which is the origin of the optimal waveform variability? Is it related only to the LD model? Or also to the optical setup and configuration? Is there variability between LD devices with the same model?

Author Response

We thank the reviewer for his careful work. In the attached file we reported all the responses.

Reviewer 2 Report

Laser self-mixing has been studied as a promising measurement technology in laboratory environment for many years, but it has not been widely implemented as commercial instruments. In this paper, the authors put some efforts to promote this technology into commercial rangefinder by applying multiple modulation of laser frequency. Although this work is still an attempt, it is very cheering to see the success of this technology in commercial products in the future. In my opinion, this paper merits for publishing in Sensors. Just some minor concerns as below:

1. Does the proposed method have limits of the triangular modulation frequencies? Does it require high modulation frequency?

2. In Section 5.2, authors proposed to measure vibration. How about the differences between distance and vibration measurement? Does it have limits for vibration frequency and amplitude?

3.    It is better to keep the unit same in different figures, e.g., time in Figure 4, 5 and 6.

Author Response

We thank the reviewer for his comments.

In the attached file we reported all the responses.
